# Growth of ocean thermal energy conversion resources under greenhouse warming regulated by oceanic eddies

**Tianshi Du[1,2], Zhao Jing [1,2] ✉, Lixin Wu [1,2], Hong Wang [1,2], Zhaohui Chen [1,2], Xiaohui Ma [1,2], Bolan Gan [1,2] & Haiyuan Yang[1,2]**

The concept of utilizing a large temperature difference (>20 °C) between the surface and deep seawater to generate electricity, known as the ocean thermal energy conversion (OTEC), provides a renewable solution to fueling our future. However, it remains poorly assessed how the OTEC resources will respond to future climate change. Here, we find that the global OTEC power potential is projected to increase by 46% around the end of this century under a high carbon emission scenario, compared to its present-day level. The augmented OTEC power potential due to the rising sea surface temperature is partially offset by the deep ocean warming. The offsetting effect is more evident in the Atlantic Ocean than Pacific and Indian Oceans. This is mainly attributed to the weakening of mesoscale eddy-induced upward heat transport, suggesting an important role of mesoscale eddies in regulating the response of thermal stratification and OTEC power potential to greenhouse warming.

Fossil fuels as energy sources have been on heavy dependence since pre-industrial times, resulting in massive emissions of greenhouse gases, especially carbon dioxide ($CO_2$)[1]. Global warming and ocean acidification caused by the increased concentration of $CO_2$ exert many different adverse effects on the ecosystems and human society[2,3]. Therefore, the replacement of fossil energy with decarbonized sources is essential for tackling these crises[4,5]. Unlike many other renewable technologies based on intermittent energy sources such as winds and sunlight[6,7], the ocean thermal energy conversion (OTEC) is capable of steadily providing humanity with vast amounts of electrical power[1,8] associated with a moderate levelized cost of energy (140–157 USD MWh$^{-1}$)[7] and competitive by-products[9]. It has been estimated that the total electrical power generated by OTEC (also termed as the OTEC power potential for short) across the global ocean could reach up to 8–10 terawatts (TW)[10], with only 1.9 TW of electricity generation by fossil fuels during 2020 in contrast[11], indicating an abundant energy potential.

The OTEC power potential density $P_{net}$ (i.e., the OTEC power potential per unit area) depends on the squared temperature difference $\Delta T$ between surface and deep (1000 m) seawater (see 'OTEC power potential density' in Methods section). During the past half century, more than 90% of the anthropogenic heat surplus accumulated in the oceans via the heat flux at the sea surface, of which two thirds is absorbed in the upper-700 m water column[12]. Accordingly, the global mean sea surface temperature (SST) has increased by an amount (0.78 °C)[13] larger than the deep ocean counterpart (0.06 °C averaged within 700–2000 m) since 1960s[14], leading to enhanced thermal stratification[15]. In the future, the strengthening of thermal stratification is likely to continue due to greenhouse warming[16], implying enriched OTEC resources. Despite this simple response of thermal stratification to greenhouse warming in terms of the global average, the geographic pattern of anthropogenic thermal stratification change is complicated[15] and strongly affected by the heat transport of oceanic flows[17–20]. On the one hand, the SST changes caused by local sea surface heat flux changes can be advected elsewhere by oceanic flows like a passive tracer, particularly into the deep ocean via the ventilation processes[19]. On the other hand, changes in surface wind and buoyancy forcing under greenhouse warming drive changes in

[1]Frontiers Science Center for Deep Ocean Multispheres and Earth System and Key Laboratory of Physical Oceanography, Ocean University of China, Qingdao, China. [2]Laoshan Laboratory, Qingdao, China. ✉e-mail: jingzhao@ouc.edu.cn

oceanic flows that redistribute the heat in the ocean and further affect the efficiency of ocean uptake of anthropogenic heat surplus via the redistribution feedback[20]. Yet existing knowledge of OTEC power potential change in a warming climate is mainly derived from a one-dimensional (1-D) model[21] with an oversimplified representation of heat transport by oceanic flows[8,22]. This causes large uncertainties in the projected OTEC power potential change by the 1-D model.

Coupled global climate models (CGCMs) provide a more reliable projection of ocean thermal stratification change under greenhouse warming compared to the 1-D model. However, so far most of the CGCMs participating in the Coupled Model Intercomparison Project Phase 6 (CMIP6)[23] have an oceanic resolution (-1°) too coarse to resolve the ocean thermal structure in the coastal ocean where deploying OTEC plants is much more feasible than in the open ocean[24]. Moreover, even in the open ocean, coarse-resolution CGCMs have an evident bias in the simulated ocean thermal stratification[25,26] partially due to deficiencies in representing effects of smaller-scale processes such as oceanic mesoscale eddies[27–29], casting doubt on their validity in projecting the trends of ocean thermal stratification and OTEC power potential in the future.

In this study, we use an unprecedented long-term high-resolution simulation based on the Community Earth System Model[30] (referred to as CESM-H) (see 'CESM-H simulation' in Methods section) to assess the change of OTEC power potential and its underlying dynamics under the high carbon emission scenario. It should be noted that the assessment in this study does not consider the feedback effect on ocean thermal stratification caused by the effluent discharge when utilizing OTEC[31]. Nevertheless, it has been demonstrated[8,22] that such feedback is negligible for a moderate cold-water intake rate such as 5 m year$^{-1}$ used for the computation of OTEC power potential in this study (see 'OTEC power potential density' in Methods section for details). As shown below, CESM-H shows good skills in reproducing the OTEC power potential in the past half century, providing us confidence in its reliability in projecting the future change of OTEC power potential.

## Results

### Simulated OTEC power potential in the historical period by the CESM-H

Performance of the CESM-H in simulating the OTEC power potential is validated against a global long-term ocean heat content (OHC) observation (see 'Observation products' in Methods section) and compared to an ensemble of coarse-resolution (-1°) CGCMs in CMIP6 (Supplementary Table 1). The time series of global OTEC power potential in the CESM-H and observation agree reasonably well with each other in terms of their time-mean values and secular changes (Fig. 1a). On the one hand, the time-mean global OTEC power potential during 1955–2021 is 8.55 ± 0.11 TW in the CESM-H (see 'Computation of standard error' in Methods), differing by less than 10% from its observational counterpart 9.36 ± 0.04 TW. On the other hand, the linear trend of global OTEC power potential simulated by the CESM-H is 1.99 ± 0.59 TW century$^{-1}$ during 1955–2021, not significantly different from the observed value 1.80 ± 0.22 TW century$^{-1}$. In contrast, the time-mean global OTEC power potential for the CMIP6 CGCM ensemble mean is 7.21 ± 0.30 TW during 1955–2021, indicating a smaller time-mean $\Delta T$ or equivalently weaker thermal stratification in the CMIP6 CGCMs than the observation and CESM-H (Supplementary Fig. 1a). This might be partially due to deficiencies of parameterizations used in coarse-resolution CGCMs for representing the restratification effect by oceanic mesoscale eddies[27–29]. Despite a smaller time-mean global OTEC power potential, the CMIP6 CGCM ensemble mean simulates a linear trend of global OTEC power potential (2.08 ± 0.31 TW century$^{-1}$) during 1955–2021 similar to those in the observation and CESM-H, in accordance with the similarity among the linear trends of $\Delta T$ in the

observation, CESM-H and CMIP6 CGCM ensemble mean (Supplementary Fig. 1a).

We next examine the present-day (1992–2021) spatial distribution of $P_{net}$ (Fig. 1b–d) that is closely related to the variability of $\Delta T$ in space. In the observation, the value of $\Delta T$ generally ranges from 0°C to 25°C in the global ocean (Supplementary Fig. 2c), making the OTEC only available over approximately half of the global ocean (Fig. 1b). The spatial distributions of $\Delta T$ and $P_{net}$ are primarily attributed to that of the SST (Supplementary Fig. 2a). As the SST decreases poleward due to the latitudinally varying solar radiation, a nonzero $P_{net}$ is mainly confined to the low-latitude regions between 35°S–40°N. Furthermore, there is a notable zonal asymmetry in the $P_{net}$. In the tropics, the SST and $P_{net}$ are higher in the Indo-Pacific warm pool than the Pacific and Atlantic equatorial cold tongues. The former is due to the accumulation of warm surface water by the wind-driven ocean circulations, whereas the latter is due to the upwelling of cold water from the thermocline into the surface layer[32]. In the subtropical oceans, high values of SST and $P_{net}$ are centered in the west of ocean basins caused by the wind-driven anticyclonic ocean circulations[32]. The value of SST is further reduced in the eastern boundary upwelling systems due to the intense upwelling of cold deep water driven by along-shore equatorward winds[33], leading to zero $P_{net}$ in these regions. The deep ocean temperature at 1000 m $T_{1000}$ spatially varies to a less extent compared to the SST (Supplementary Fig. 2b) but plays a non-negligible role in the regional variability of $\Delta T$ and $P_{net}$. In particular, the injection of salty, warm Mediterranean Water into the deep Atlantic Ocean results in a high value of $T_{1000}$ in the eastern subtropical Atlantic Ocean[34,35], reducing $\Delta T$ to below 20°C and making $P_{net}$ become zero. Similarly, the lower values of $\Delta T$ and $P_{net}$ in the Arabian Sea than in the adjacent ocean are due to the injection of salty, warm Red Sea Water[36]. The above geographic features of $P_{net}$ are qualitatively reproduced by both the CESM-H and CMIP6 CGCM ensemble mean (Fig. 1b–d). However, the CESM-H outperforms the CMIP6 CGCM ensemble mean quantitatively. For instance, the time-mean (1955–2021) area of the global OTEC region, defined as the region with nonzero $P_{net}$ (see 'OTEC power potential density' in Methods for more details), is 127.4 ± 0.3 million km$^2$ in the observation, closer to 116.6 ± 1.0 million km$^2$ in the CESM-H than 99.9 ± 3.2 million km$^2$ in the CMIP6 CGCM ensemble mean.

At present and in the foreseen future, deployment of OTEC plants in the exclusive economic zone (EEZ) would be more practical and cost-effective[24]. The EEZ covers 49.7% of the OTEC region and accounts for 52.0% of the global OTEC power potential in the observation (Fig. 1e, f), with the archipelago in the Pacific Ocean making a major contribution. The CESM-H reproduces the observed OTEC power potential in the EEZ reasonably well (Fig. 1e, g). The time-mean OTEC power potentials in the EEZ between the CESM-H (4.69 ± 0.04 TW) and observation (4.87 ± 0.02 TW) differ by less than 5% during 1955–2021. As a result of greenhouse warming, the observed OTEC power potential in the EEZ exhibits a positive trend during 1955–2021 with a slope of 0.80 ± 0.09 TW century$^{-1}$ (Fig. 1e). The CESM-H reproduces this trend, although the simulated slope (0.97 ± 0.20 TW century$^{-1}$) is slightly higher than the observed one. Similar to the case of global OTEC power potential, the time-mean OTEC power potential over the EEZ in the CMIP6 CGCM ensemble mean (3.71 ± 0.13 TW) biases low by 23.8% (20.9%) compared to the observation (CESM-H) due to its smaller time-mean $\Delta T$ (Fig. 1e, h and Supplementary Fig. 1b). However, its simulated linear trend (0.91 ± 0.14 TW century$^{-1}$) is not statistically different from those in the observation and CESM-H.

Based on the above comparisons, we conclude that the CESM-H generally provides a reliable simulation of OTEC power potential during 1955–2021. Specifically, it simulates a linear trend of OTEC power potential similar to those in the observation and CMIP6 CGCM ensemble mean but outperforms the CMIP6 CGCM ensemble mean in the simulated time-mean OTEC power potential. This lends support to

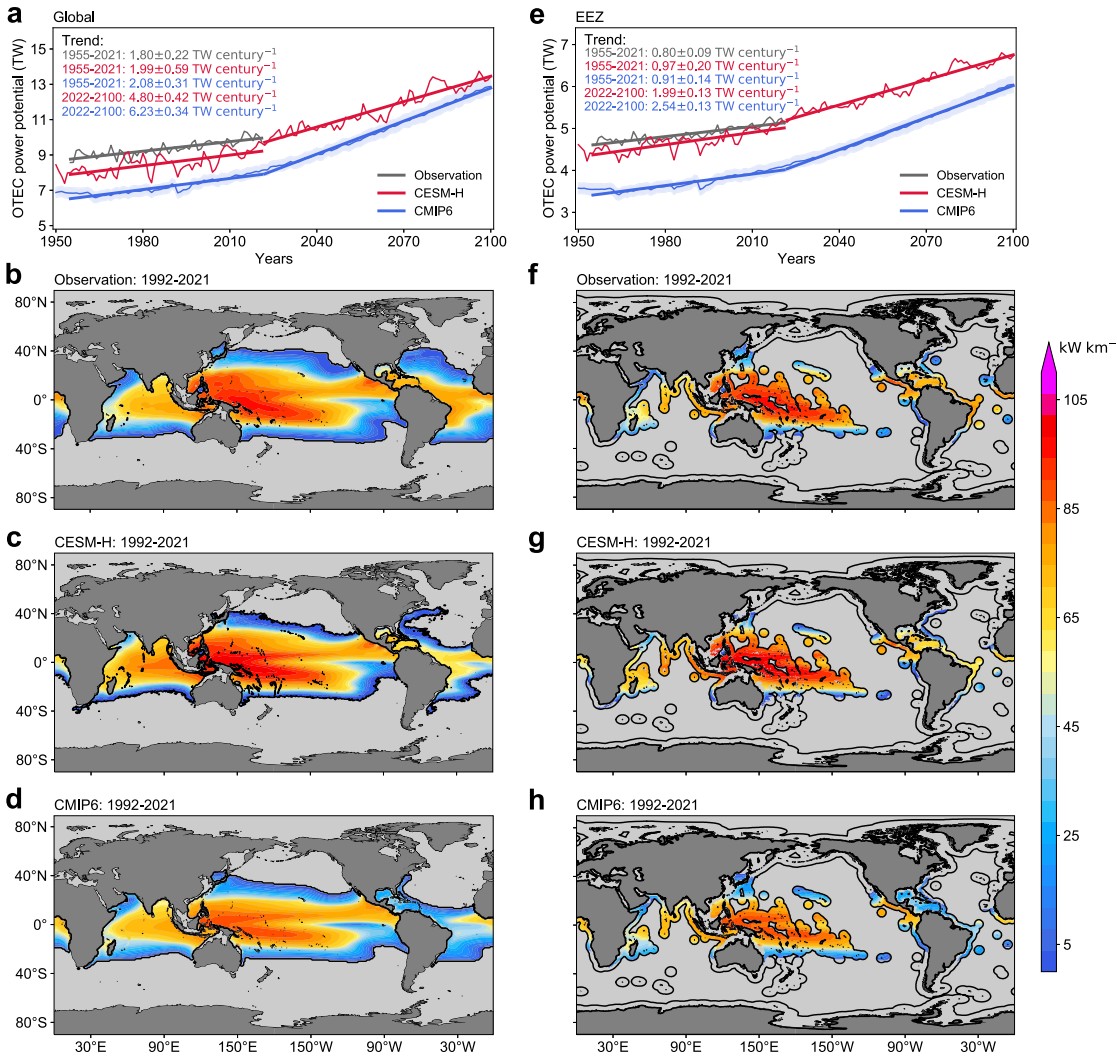

**Fig. 1 | Ocean thermal energy conversion (OTEC) resources in the observation and climate model simulations. a** Global OTEC power potential derived from the observation (gray), high-resolution Community Earth System Model (CESM-H) (red), and ensemble mean of the coupled global climate models (CGCMs) in the Coupled Model Intercomparison Project Phase 6 (CMIP6) (blue). The shading corresponds to the standard error of the CMIP6 CGCM ensemble mean. The numbers on the top left corner show the slope of linear trends of global OTEC power potential over different periods along with its standard error. **b**–**d** Geographic distribution of time-mean (1992–2021) OTEC power potential density in the observation, CESM-H and CMIP6 CGCM ensemble mean, respectively. **e**–**h** Same as **a**–**d** but for the OTEC resources within the exclusive economic zone (EEZ). The black solid lines encompass the EEZ across the globe.

using the CESM-H for projecting the future OTEC power potential change by the end of this century.

## Projected change of OTEC power potential under a high carbon emission scenario by the CESM-H

The geographic distribution of time-mean (2071–2100) $P_{net}$ around the end of this century is similar to the present one but its magnitude becomes systematically larger (Fig. 2a, b) owing to the enhanced ocean thermal stratification under greenhouse warming. The projected linear trend of global OTEC power potential during 2022–2100 ($4.80 \pm 0.42$ TW century⁻¹) is more than twice $1.99 \pm 0.59$ TW century⁻¹ during 1955–2021 (Fig. 1a). The time-mean global OTEC power potential during 2071–2100 will increase to $12.88 \pm 0.18$ TW, about 45.5% larger than its present-day (1992–2021) level $8.85 \pm 0.12$ TW. The increased global OTEC power potential is contributed by both an expanded OTEC region and an augmented $P_{net}$ (Supplementary Fig. 3). The former increases from $119.1 \pm 0.9$ million km² during 1992–2021 to $149.0 \pm 0.8$ million km² during 2071–2100, while the latter increases from $74.3 \pm 0.6$ kW km⁻² to $86.4 \pm 0.7$ kW km⁻². The increased $P_{net}$ is most prominent in the margin of the present-day OTEC region where the

frequency of $\Delta T$ exceeding 20 °C is less than 50% (see 'OTEC power potential density' in Methods for more details; Supplementary Fig. 4). This is expected because $P_{net}$ is a discontinuous function of $\Delta T$, being zero for $\Delta T < 20$ °C. Currently, the margin of the OTEC region seldom satisfies the 20 °C threshold for $\Delta T$, making the time-mean $P_{net}$ close to zero (Fig. 1c and Supplementary Fig. 4). During 2071–2100, the $\Delta T$ threshold in these marginal regions is permanently exceeded, causing a considerable increase in $P_{net}$. As to the EEZ, the trend of OTEC power potential during 2022–2100 ($1.99 \pm 0.13$ TW century⁻¹) is about twice $0.97 \pm 0.20$ TW century⁻¹ during 1955–2021 (Fig. 1e). The time-mean OTEC power potential within the EEZ during 2071–2100 is projected to increase to $6.50 \pm 0.05$ TW, 34.0% higher than its present-day level $4.85 \pm 0.03$ TW. This increase is mainly ascribed to the rising $P_{net}$ from $77.6 \pm 0.5$ kW km⁻² to $92.4 \pm 0.6$ kW km⁻², whereas the area of the OTEC region in the EEZ expands only by 12.6%.

We then focus on the more geographically restrictive regions like the South China Sea (SCS) and Gulf of Mexico (GOM), the two representative marginal seas of the Pacific and Atlantic Oceans, respectively. The SCS located at lower latitudes has generally higher SST than the GOM, resulting in larger values of present-day $\Delta T$ and $P_{net}$

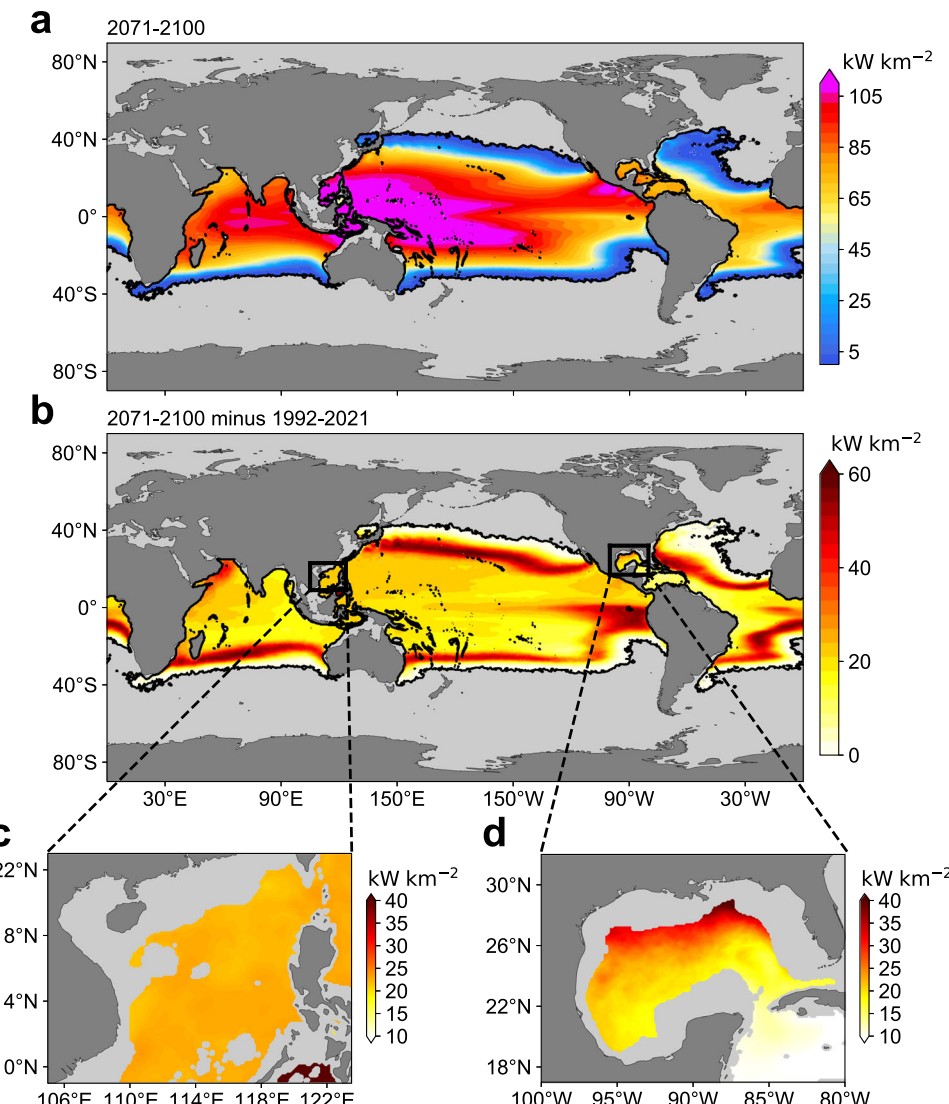

**Fig. 2 | Projected change of ocean thermal energy conversion (OTEC) resources under a high carbon emission scenario by the high-resolution Community Earth System Model (CESM-H).** Geographic distribution of time-mean OTEC power potential density during 2071–2100 in the CESM-H (**a**) and its difference from the time-mean value during 1992–2021 (**b**). **c**, **d** Same as **b**, but for the zoomed-in plots of the South China Sea and the Gulf of Mexico.

(Supplementary Fig. 2 and Fig. 1b). In the future, both the SCS and GOM exhibit an evident increase in $P_{net}$ by the end of this century. But their spatial structures of $P_{net}$ changes differ substantially (Fig. 2c, d). The change in $P_{net}$ from 1992–2021 to 2071–2100 is relatively homogenous in the SCS, being 23 kW km$^{-2}$ or so. In contrast, the change of $P_{net}$ in the GOM is larger in the northern part (-40 kW km$^{-2}$) but decreases southeastward to -15 kW km$^{-2}$ near the Yucatan Channel. This heterogeneous change of $P_{net}$ mimics that of SST change under greenhouse warming (Supplementary Fig. 5). The depressed SST increase in the southeastern GOM is likely caused by the weakened Loop Current associated with a decline of the Atlantic Meridional Overturning Circulation under greenhouse warming[37] that reduces the warm water intrusion into the GOM via the Yucatan Channel[38].

## Increase of OTEC power potential under greenhouse warming retarded by deep ocean warming

As $P_{net}$ depends on the value of $\Delta T$, sea surface warming acts to increase $P_{net}$ but deep ocean warming has the opposite effect. Although the surface ocean is projected to warm faster than the deep ocean in terms of the global average[16] (Supplementary Fig. 6a), the local change of $\Delta T$ under greenhouse warming is controlled by

complicated dynamics and may differ substantially from its global mean value. To assess the effect of deep ocean warming on the projected OTEC power potential trend in different regions, we recompute the OTEC power potential density by fixing $T_{1000}$ as its time-mean value during 1992–2021 (denoted as $P_{net}^{fix}$). The linear trend of global OTEC power potential derived from $P_{net}^{fix}$ (denoted as the OTEC-Fix power potential) during 2022-2100 is 5.95 ± 0.44 TW century$^{-1}$, 24.0% larger than 4.80 ± 0.42 TW century$^{-1}$ derived from $P_{net}$ (Fig. 3a). As to the OTEC power potential density and area of the OTEC region, neglecting the effect of deep ocean warming overestimates their linear trends by 22.9% and 21.4%, respectively (Supplementary Fig. 3). This offsetting effect is spatially inhomogeneous due to the varying deep ocean warming rate across the OTEC region (Supplementary Fig. 6b). It is more important in the Atlantic Ocean where the linear trend of OTEC-Fix power potential during 2022-2100 is 54.3% greater than that of OTEC power potential (Fig. 3b). In contrast, the linear trends of OTEC-Fix and OTEC power potentials differ only by 15.8% and 22.8% in the Pacific and Indian Oceans, respectively (Fig. 3c, d). The difference between the effects of deep ocean warming on the OTEC power potential change in the Pacific and Atlantic Oceans becomes even more evident for their marginal seas, i.e., the SCS and GOM (Fig. 3e, f).

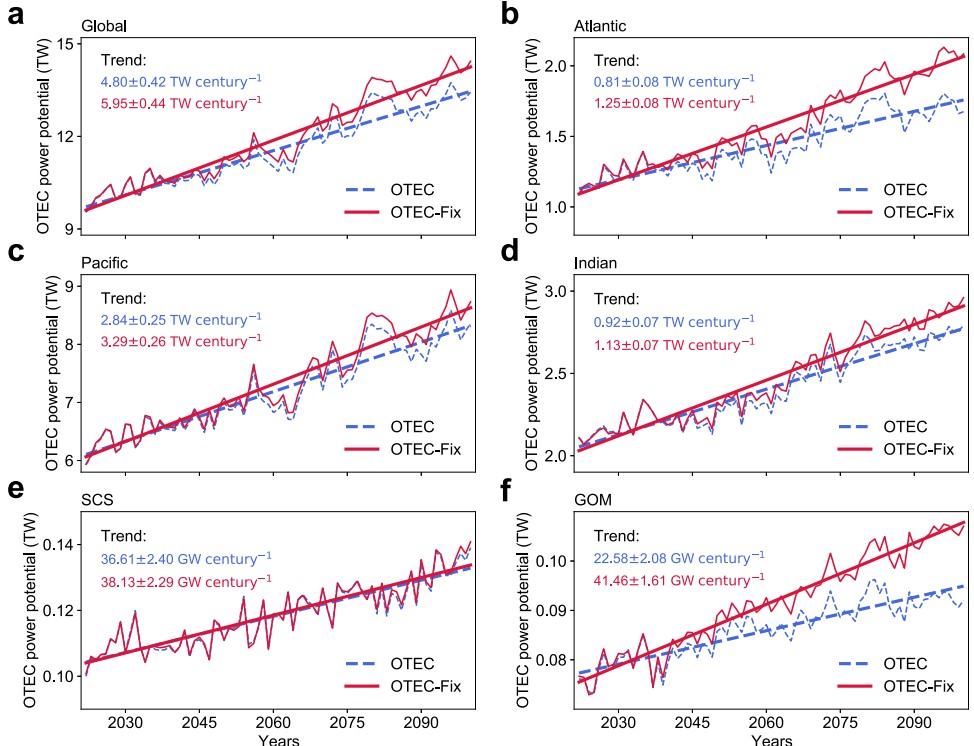

**Fig. 3 | Offsetting effect of deep ocean warming on the ocean thermal energy conversion (OTEC) resource increase under a high carbon emission scenario derived from the high-resolution Community Earth System Model (CESM-H).**
**a** Time series of global OTEC power potential during 2022–2100 (blue dashed line) and its counterpart (OTEC-Fix power potential) computed by fixing the temperature at 1000 m as its time-mean value during 1992–2021 (red line). Slope of the linear trends along with its standard error is shown in the top left corner. **b–f** Same as **a**, but for the OTEC power potential and OTEC-Fix power potential in the Atlantic, Pacific, Indian Oceans, South China Sea (SCS), and Gulf of Mexico (GOM), respectively. Domain of different oceans is labeled in Supplementary Fig. 6b.

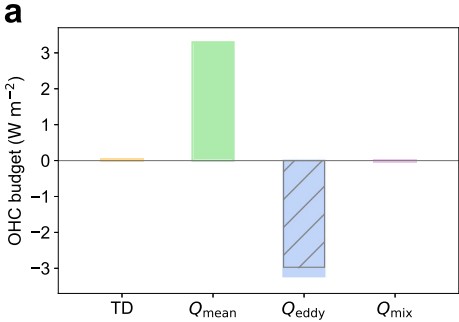
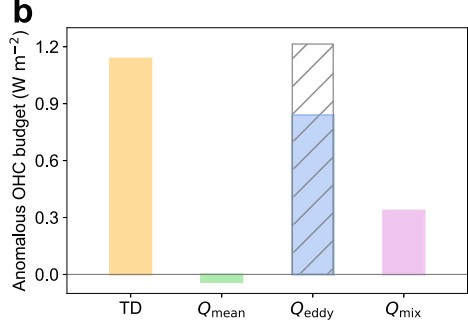

**Fig. 4 | Deep ocean warming in the Atlantic Ocean due to the weakened upward heat transport by mesoscale eddies. a** The ocean heat content (OHC) budget in the 800–1200 m water column over the Atlantic ocean thermal energy conversion (OTEC) region (Supplementary Fig. 6b) during 1992–2021 where TD represents the OHC tendency, $Q_{mean}$ the heat transport convergence by mean flows, $Q_{eddy}$ the heat transport convergence by mesoscale eddies with the contribution of vertical eddy heat transport at 800 m marked by hatched lines, and $Q_{mix}$ the parameterized turbulent vertical mixing. **b** Same as **a**, but for the anomalous OHC budget under greenhouse warming (i.e., 2071–2100 minus 1992–2021).

Specifically, the linear trends of OTEC-Fix and OTEC power potentials are almost the same in the SCS, whereas the linear trend of OTEC-Fix power potential is 83.6% greater than that of OTEC power potential in the GOM.

To uncover the underlying dynamics of strong warming at 1000 m in the Atlantic Ocean, an OHC budget analysis is performed for the 800–1200 m water column over the Atlantic OTEC region (see 'OHC budget analysis' in Methods; Supplementary Fig. 6b). Changing the range of the water column for analysis to 900–1100 m or 600–1400 m does not have any substantial impact on the following results (Supplementary Fig. 7). For the climatological mean OHC budget during 1992–2021 (Fig. 4a), there is an approximate balance between the heat supply caused by mean flows ($3.29 \pm 0.18$ W m$^{-2}$) and heat sink by mesoscale eddies ($-3.22 \pm 0.09$ W m$^{-2}$), with the turbulent vertical mixing ($-0.03 \pm 0.01$ W m$^{-2}$) and OHC tendency ($0.05 \pm 0.14$ W m$^{-2}$) more than an order of magnitude smaller. The cooling by mesoscale eddies is largely attributed to their induced upward heat transport at 800 m ($-2.97 \pm 0.10$ W m$^{-2}$) as a result of baroclinic instability[27,39].

As to the anomalous OHC budget under greenhouse warming (i.e., 2071–2100 minus 1992–2021), the OHC tendency anomaly ($1.13 \pm 0.09$ W m$^{-2}$) becomes one of the dominant terms (Fig. 4b). This OHC tendency anomaly is primarily attributed to mesoscale eddies ($0.84 \pm 0.05$ W m$^{-2}$). They account for 74.1% of the OHC tendency anomaly, whereas this fraction value is reduced to -3.4% and 29.3% for

mean flows and turbulent vertical mixing, respectively. The effect of mesoscale eddies is mostly ascribed to the reduction of upward eddy heat transport at 800 m. Its value decreases from $2.97 \pm 0.10$ W m$^{-2}$ during 1992–2021 to $1.76 \pm 0.04$ W m$^{-2}$ during 2071–2100. Such a decrease implies a weakened baroclinic instability under greenhouse warming which might be partially due to the enhanced stratification[16,40,41] that reduces the available potential energy stored in mean flows through flattening the isopycnals[26,42]. It should be noted that the reduced upward heat transport of mesoscale eddies also occurs in the Pacific and Indian Oceans. However, the upward eddy heat transport in the Pacific and Indian Oceans has a much shallower vertical structure than that in the Atlantic Ocean (Supplementary Fig. 8) possibly due to the shallower thermocline in the Pacific and Indian Oceans[32,43], contributing negligibly to the OHC tendency and its anomaly between 800 and 1200 m.

## Discussion

Our study provides an assessment of OTEC resources change in the future. The OTEC power potential is projected to become more abundant in response to greenhouse warming, including an expanded OTEC region and increased OTEC power potential density. The change of OTEC power potential under greenhouse warming is not simply determined by the change of SST but can be locally strongly affected by the deep ocean temperature change, necessitating a better understanding of the deep ocean's response to greenhouse warming for a more accurate assessment of OTEC power potential change in the future. Additionally, vertical heat transport by mesoscale eddies is found to play an important role in regulating the response of OTEC power potential to greenhouse warming via its contribution to deep ocean warming. This heat transport cannot be accommodated in the simple 1-D model used in the previous analysis[21]. Neither is it well represented in the coarse-resolution CGCMs, given the deficiencies in the parameterizations of mesoscale eddy heat transport[27–29]. These deficiencies may be partially responsible for the overly small climatological mean OTEC power potential in the coarse-resolution CMIP6 CGCMs. Furthermore, although the linear trends of OTEC power potential between the CESM-H and coarse-resolution CMIP6 CGCMs do not statistically differ from each other during the historical period (1955–2021), it does not necessarily mean that the OTEC power potentials in the CESM-H and coarse-resolution CMIP6 CGCMs have consistent responses to greenhouse warming. In fact, with the rising greenhouse gas emission in the future as in the high carbon emission scenario, the difference in the projected trends of OTEC power potential between the CESM-H and coarse-resolution CMIP6 CGCMs during 2022-2100 is qualitatively similar to its historical counterpart but quantitatively becomes sufficiently large to be statistically significant (Fig. 1a, e). It remains unclear to what extent the change of mesoscale eddy heat transport under greenhouse warming can be reproduced by the parameterizations, and to what extent the potential difference between the resolved and parameterized mesoscale eddy heat transport changes may affect the response of OTEC power potential to greenhouse warming. These uncertainties can be circumvented by using high-resolution CGCMs resolving mesoscale eddies to evaluate the OTEC power potential change in the future. Finally, there is a caveat that the CESM-H, as well as CMIP6 CGCMs, does not simulate the feedback effect on ocean thermal stratification caused by the effluent discharge when utilizing OTEC[31]. This limits us to assign a moderate cold-water intake rate under which condition the feedback is negligible[8,22]. The amount of global OTEC power potential at present and its future increase would be even larger if a greater intake rate were used for the computation of OTEC power potential. In this sense, the values reported in this study should be treated as a conservative estimate of OTEC power potential change under the high carbon emission scenario.

## Methods

### CESM-H simulation

The CESM-H simulation is performed based on CESM version 1.3. It has a nominal 0.25° and 0.1° horizontal resolution for the atmosphere and ocean components, respectively. There are 30 vertical levels in the atmosphere and 62 levels in the ocean. The simulation consists of two components. The first one is the 500-year-long pre-industrial control (PI-CTRL) simulation with the climate forcings fixed to the 1850 conditions. The second one is the 250-year-long historical and future transient simulation (HF-TNST) during the 1850–2100 period. The HF-TNST simulation is branched off from the 250[th] year of PI-CTRL and forced by the historical forcing from 1850 to 2005 and concentration pathway 8.5 (RCP8.5) forcing during 2006–2100. Comprehensive descriptions of CESM-H can be found in a recent overview paper[30].

Monthly averaged potential temperature and diagnostic outputs of heat transport by resolved flows are saved during 1878-2100. In this study, we use the potential temperature as a proxy for temperature. This may introduce a small error on the order of 0.1 °C for the temperature at 1000 m[44]. This error is two orders of magnitude smaller than the $\Delta T$ threshold (20 °C) for OTEC and does not affect the secular change of OTEC power potential.

After 250 year's spin-up, the model drift in the deep ocean (1000 m) temperature becomes small but still noticeable[30]. This may bias the simulated secular change of OTEC power potential in HF-TNST caused by greenhouse warming. To minimize such bias, we subtract the trend of temperature at 1000 m during the years 350–500 in PI-CTRL from that during 1950–2100 in HF-TNST.

### Observation products

An observational global OHC dataset provided by Japan Meteorological Agency (JMA)[45] is used in this study to evaluate the OTEC resources during 1955–2021. The monthly temperature data are on 1° × 1° regular grids and have 26 levels between 0–2000 m.

### OTEC power potential density

The OTEC power potential density is defined as the net electrical power generated by the closed-cycle OTEC system[6,46,47] per unit area, formulated as[10,22,31]:

$$P_{\text{net}} = H(\Delta T - 20°\text{C})\left(w_{\text{cw}}\frac{3\rho c_p \varepsilon_{\text{tg}}\gamma}{16(1+\gamma)}\frac{(\Delta T)^2}{T} - P_{\text{pump}}\right) \qquad (1)$$

$$P_{\text{pump}} = w_{\text{cw}}0.3\frac{\rho c_p \varepsilon_{\text{tg}}\gamma}{4(1+\gamma)} \qquad (2)$$

where H represents a Heaviside function being 1 when its argument is positive and being zero otherwise, $w_{\text{cw}} = 5$ m year$^{-1}$ is the deep cold seawater intake rate, $\rho = 1026$ kg m$^{-3}$ is a reference density of seawater, $c_p = 4000$ J (kg·K)$^{-1}$ is the seawater specific heat capacity, $\varepsilon_{\text{tg}} = 0.75$ is the turbo-generator combined efficiency, $\gamma$ is the flow rate ratio of the surface warm water intake over the deep cold water fixed as 1.5, $T$ is the absolute temperature of the warm water intake and $\Delta T$ is the temperature difference between surface and deep ocean (1000 m) seawater intakes[48]. The first term on the right-hand side of Eq. (1) represents the gross power density $P_{\text{g}}$ of the OTEC system. Besides, a power consumption $P_{\text{pump}}$ when operating the OTEC system is included as the second term on the right-hand side. It corresponds to 30% of $P_{\text{g}}$ at a design condition of $\Delta T = 20$ °C and $T = 300$ K[31,49].

According to Eq. (1), a value of $\Delta T$ larger than 20 °C is required to yield nonzero $P_{\text{net}}$. We define the region where the value of $\Delta T$ exceeds 20 °C as the OTEC region. It should be noted that the OTEC region as well as its area varies with time due to the temporal variation of $\Delta T$. Accordingly, the OTEC region marked in the global map (e.g., Fig. 1b and Supplementary Fig. 6b) for a given period is defined as the union of all the instantaneous OTEC regions during that period or

equivalently the domain where the maximum of $\Delta T$ during that period exceeds 20 °C. The margin of the OTEC region for a given period is defined as the domain where the frequency of $\Delta T$ exceeding 20 °C is less than 50% during that period (e.g., domain between gray and black contours in Supplementary Fig. 4).

It should be noted that the cold water used in the system is pumped up from the deep ocean and will not be discharged back to its initial depth, but directly to the surface, leading to a reduction in the SST, $\Delta T$, and subsequently $P_{net}$[31,50]. However, previous studies[8,22] have revealed that with $w_{cw}$ smaller than 5 m year$^{-1}$, $P_{net}$ is linearly proportional to $w_{cw}$ as if there were no feedback effect. Under this condition, the impact of utilizing OTEC on the ocean thermal stratification in practice has been proved ignorable even on a thousand-year time scale. The $w_{cw}$ of 5 m year$^{-1}$ is on the same order of the large-scale vertical velocity in the mid-depth ocean interior estimated to be O(1 m year$^{-1}$)[51,52] but is more than an order of magnitude smaller than the intense wind-driven upwelling O(1 m day$^{-1}$) in the eastern boundary upwelling systems[53].

### Computation of standard error

To compute the standard error (s.e.) of statistics discussed in the main text such as the time-mean value and slope of the linear trend, we model the time series of some quantity $\theta(t)$ as:

$$\theta(t) = \beta_1 t + \beta_0 + \tau(t) \tag{3}$$

where $\beta_1 t$ is a deterministic linear trend with $\beta_1$ being its slope, $\beta_0$ is an intercept and $\tau(t)$ is a stationary stochastic process with zero mean. The origin of the time axis is set as the center of time period for analysis, in which case the time-mean $\theta(t)$ over that period is equal to $\beta_0$. The values of $\beta_1$ and $\beta_0$ are estimated using the ordinary least square (OLS). Due to the autocorrelation in $\tau(t)$, the Cochrane-Orcutt procedure[54] is used to compute the s.e. of OLS estimators $\hat{\beta}_1$ and $\hat{\beta}_0$ (denoted as s.e.$(\hat{\beta}_1)_{OLS}$ and s.e.$(\hat{\beta}_0)_{OLS}$) with the premise that $\tau(t)$ can be approximated as a first-order autoregressive process.

For the coarse-resolution CMIP6 CGCMs, there is additional statistical uncertainty from estimating its ensemble mean. Such uncertainty is quantified as the s.e. of the estimators of the ensemble mean:

$$\text{s.e.}(\{\hat{\beta}_1\})_{EM} = \frac{1}{\sqrt{N}}\sqrt{\frac{1}{N-1}\sum_{i=1}^{N}(\hat{\beta}_{1,i} - \{\hat{\beta}_1\})^2} \tag{4}$$

$$\text{s.e.}(\{\hat{\beta}_0\})_{EM} = \frac{1}{\sqrt{N}}\sqrt{\frac{1}{N-1}\sum_{i=1}^{N}(\hat{\beta}_{0,i} - \{\hat{\beta}_0\})^2} \tag{5}$$

where $N = 36$ is the ensemble number of coarse-resolution CMIP6 CGCMs, the subscript $i$ corresponds the $i$-th CGCM, and the braces represent the average across the CGCMs. To derive Eqs. (4) and (5), we assume the independence among CGCMs. The s.e. of $\{\hat{\beta}_1\}$ is defined as the larger one between s.e.$(\{\hat{\beta}_1\})_{EM}$ and s.e.$(\{\hat{\beta}_1\})_{OLS}$. So is the case for the s.e. of $\{\hat{\beta}_0\}$.

### OHC budget analysis

The OHC budget for the 800-1200 m water column over the Atlantic OTEC region is derived as:

$$\left\langle \int_{z_l}^{z_u} \rho_0 c_p \frac{\overline{\partial T}}{\partial t} dz \right\rangle = -\left\langle \int_{z_l}^{z_u} \rho_0 c_p \nabla \cdot (\overline{\mathbf{u}}\overline{T}) dz \right\rangle - \left\langle \int_{z_l}^{z_u} \rho_0 c_p \nabla \cdot (\overline{\mathbf{u}'T'}) dz \right\rangle + \left\langle \int_{z_l}^{z_u} \rho_0 c_p \frac{\partial}{\partial z}\left(\overline{K_m \frac{\partial T}{\partial z}}\right) dz \right\rangle \tag{6}$$

where $z_u$ = -800 m, $z_l$ = -1200 m, $\mathbf{u} = (u, v, w)$ is the three-dimensional oceanic flow, $K_m$ is the turbulent vertical diffusivity, $\nabla = (\partial/\partial x, \partial/\partial y,$

$\partial/\partial z)$, the overbar denotes the three-month average, the prime denotes the perturbation from the three-month average, and $\langle ... \rangle$ denotes the horizontal average. The horizontal mixing by subgrid-scale processes is dropped, as its effect is negligible compared to other terms when averaged over a sufficiently large area considered here, i.e., the Atlantic OTEC region (Supplementary Fig. 6b; Supplementary Fig. 9). The mean flow signals are isolated through the three-month average, while the mesoscale eddy field is defined as the perturbation.

The term on the left-hand side of Eq. (6) is the OHC tendency (TD) that is output by the CESM-H. The terms on the right-hand side in sequence are the heat transport convergence of mean flows ($Q_{mean}$), the heat flux convergence of mesoscale eddies ($Q_{eddy}$), and turbulent vertical mixing ($Q_{mix}$). The $Q_{mean}$ is computed based on the monthly model output of $\mathbf{u}$ and $T$, while the $Q_{eddy}$ is derived from subtracting $Q_{mean}$ from the model diagnostic output of $\mathbf{u}T$. The $Q_{mix}$ is computed as the residue.

## Data availability

All data needed to evaluate the conclusions in the paper can be downloaded from the following links: CESM-H, https://ihesp.tamu.edu, and http://ihesp.qnlm.ac; CMIP6 CGCMs, https://esgf-node.llnl.gov/search/cmip6/; JMA, https://www.data.jma.go.jp/gmd/kaiyou/english/ohc/ohc_global_en.html; Extraction of EEZ regions, https://www.marineregions.org/eezsearch.php.

## Code availability

The iHESP version of the CESM-H code is available at ZENODO via https://zenodo.org/record/3637771.

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

## Acknowledgements
Z.J. is supported by the Marine S&T Fund of Shandong Province for Laoshan Laboratory (2022LSL010302), and the Taishan Scholar Funds (tsqn201909052). We thank Laoshan Laboratory for the support of computing resources.

## Author contributions
Z.J. wrote the manuscript and instructed T.D. to perform the analysis. Z.J. and L.W. proposed the central idea of the manuscript. H.W. conducted the CESM-H simulation. Z.C., X.M., B.G. and H.Y. were involved in interpreting the results and associated dynamics.

## Competing interests
The authors declare no competing interests.
