## [Peer Review File · Nature Communications]

Growth of Ocean Thermal Energy Conversion Resources under Greenhouse Warming Regulated by Oceanic EddiesREVIEWER COMMENTS

Reviewer #1 (Remarks to the Author):

This paper addresses an issue that is extremely timely, that is whether the amount of energy that could be extracted from OTEC could potentially significantly decrease or increase in the future. They use a state of the art high resolution climate model in the analysis. The manuscript would be suitable for publication if it is significantly edited for clarity. In several places the authors overstate the differences between the results found with the high-resolution simulation and that found in CMIP6 with respect to the changes in OTEC potential in the future. It would be good to include the location of the EEZs on the one of the maps, for instance Figure 1. In addition, the authors should report the standard deviations and/or significance level for their calculations including those mentioned in my comments below.

LuAnne Thompson, School of Oceanography, University of Washington

Specific comments follow below.

Line 17: I believe that the authors mean "power potential".

Line 19: "the exclusive economic zone" needs to be modified for clarity. Maybe the authors mean the exclusive economic zones across the globe?

Line 21: In the North Atlantic?

Line 23: The authors should remove the phrase starting with "unresolved by the majority of current generation of climate models" as those models parameterize the impact of eddies the eddies. The additional factor that was not directly examined was the impact of a weaker mean flow in the control climates in the CMIP6 that could explain the offset shows in Extended Data Figure 1. In addition, Extended Data Fig. 1 shows clearly that delta T is reduced by a similar amount in the CMIP6 runs as that in CESM-H. This figure suggests that the increase in OTEC potential may be well represented by the parameterized eddies in the CMIP6 models even if the mean OTEC is not.

Line 30: Please give an example of what is meant by "intermittent renewable technologies"

Line 35: At some point in the paper, the large cost of implementation of OTEC should be mentioned when compared to other technologist.

Line 41-42: I am not sure what the authors mean by "with even a larger increasing rate"? Do they mean because global warming is expected to accelerate? Doesn't this depend on the emission scenario considered?

Line 44: More explanation is needed here: do the authors think that contributions from ocean heat transport by the large scale flow is expected to change?

Line 46: Please clarify what is meant by "more reliable"? Is this reference when compared against the 1-D models?

Line 48: I assume you men 1 degree (not 1 degree C).

Line 49: I assume the authors mean the coastal ocean here. That should be said explicitly.

Line 61: What is mean by "moderate cold-water intake"? This is not clear from the description of OTEC provided here.

Line 63: This should read "potential power" not power as the study here gives an upper limit on what could be produced.

Line 75: Please replace "only about" with the mean and standard deviation.

Line 79: the difference between the trends among the observation, CESM-H and CMIP6 appear to be small in Extended Data Fig. 1. Is this a statistically robust statement? If so, they should give the standard error in the trend estimate, including the spread across the CMIP6. Or remove this statement.

Line 89: What is "the OTEC region". Please define.

Line 98: This is a spurious comment about the coarse resolution of the observations and should be removed unless the authors do further analysis. I also suspect that error analysis would show that they are not statistically different.

Line 105: I see good agreement with the increase of the CMIP6 within the standard deviation. This sentence should be modified to reflect that even though the mean delta T is small, the increase is in relatively good agreement. This also requires that the paragraph starting at line be modified.

Line 130 the statement starting with "Zoom" is not a sentence.

Line 156: Sentence starting with "Sensitivity" does not make sense.

Line 162: Once again, it is important to note that the effects of mesoscale processes are parameterized in the CMIP6 models.

Line 175: It should be pointed out that this is likely owing to the shallower thermocline in the Pacific and Indian Oceans.

Line 182: It is important to note that while the low resolution CMIP6 simulations do not represent the mean OTEC power, the CHANGE in OTEC power is well represented.

Line 184: I do understand what is meant by this sentence.

In the figure captions "OTEC power" should be replaced with OTEC power potential.

Figure 3: the standard error of the trend estimates should be stated for each of the lines in the plots here.

Line 262: Should be changed to "the ocean thermal stratification in practice"

Line 263: how big is 5m/year compared to typical values of w from mixing or upwelling? Some context is needed. 5m/year seems like a larger number for the mid-depth ocean.

Line 274: Over what area would this hold? A few details of the sensitivity tests done should be included.

Reviewer #2 (Remarks to the Author):

The paper studies the influence of global warming on the increase in power generation of OTEC plants. Such work is useful for extracting thermal energy from the ocean. There are some concerns that the authors are encouraged to address.

1. The common range of temperature differences between the surface water and deep water in different regions of the world can be quantified in the paper. The thermocline in different regions and the reasons for differences in temperature difference with depth can be discussed to better explain the zonal differences in OTEC power. The steady power (Line 31) is mostly suitable for equatorial regions where the surface

temperatures do not vary a lot with seasons. Large temperature differences mentioned in the abstract is also not very informative since OTEC operates on temperature differences of $\sim 20^{\circ}\text{C}$.

2. It would be interesting to see in the introduction some specific global warming mechanisms that target changes in ocean water temperatures, and by how much ocean water has changed over the years. Trends in changes of ocean surface and deep water temperatures over the years would be useful - higher rate of change in temperatures is projected in Extended Fig. 1. The high carbon emission scenario leading to temperature changes of ocean water discussed in the paper can also be quantified.

3. It is easier to comprehend the influence of global warming on surface ocean water. Deep cold water used by OTEC plants is ~ 1000 m below the ocean surface and the surface atmospheric effects would be almost negligible. Also, there is a lack in natural mixing between surface warm water and deep cold water. Hence, it is not easy to follow how DOW would be directly impacted by global warming. More discussions are required, particularly on the influence of ocean currents which is a focus of this paper. DOW temperatures remain almost constant ($4 - 6^{\circ}\text{C}$) at 1000 m throughout the world, while surface temperatures vary with region. Therefore, some form of evidence is necessary that the DOW does change with global warming.

4. The different regions (Pacific, Atlantic, etc) can be labelled on the contour maps for better clarity in result presentation.

5. It is difficult to link some of the discussions in the paper with the figures. This should be improved throughout the paper. (e.g. Line 114 – 115: refers to Figure 3a and Figure 1a?; Lines 120 – 122 – not clear; margin of OTEC?). Purpose of Table 1 is not clear.

6. The geographical differences between China and Mexico affects the OTEC power due to differences in vertical ocean thermal gradients. As mentioned above, reasons for such regional differences in thermal gradients and OTEC power should be well explained.

7. As acknowledged, the OTEC power depends on the thermal gradient with depth, regardless of the rates of changes in the surface and deep ocean water. Therefore, global warming may not always improve OTEC power, since the temperature gradient could change or remain same. This could be highlighted as a key outcome and could serve as a guide for potential future researchers investigating this topic. The growth of OTEC resource mentioned in the title may not be appropriate as well.

Reply to the first reviewer

We are very grateful to you for your time in carefully reading our manuscript and providing helpful comments that make our manuscript better. We have carefully considered each of your comments (in blue) and revised the manuscript accordingly. Please find our response (in black) to your comments below.

Reviewer #1 (Remarks to the Author):

This paper addresses an issue that is extremely timely, that is whether the amount of energy that could be extracted from OTEC could potentially significantly decrease or increase in the future. They use a state of the art high resolution climate model in the analysis. The manuscript would be suitable for publication if it is significantly edited for clarity. In several places the authors overstate the differences between the results found with the high-resolution simulation and that found in CMIP6 with respect to the changes in OTEC potential in the future. It would be good to include the location of the EEZs on one of the maps, for instance Figure 1. In addition, the authors should report the standard deviations and/or significance level for their calculations including those mentioned in my comments below.

Following your comments, we have revised the statements in the manuscript to avoid unclarity. Furthermore, we have provided the standard errors for the estimated quantities and provided details on how the standard errors are computed (See “Computation of standard errors” in Methods). Based on the standard errors, we demonstrate that the time-mean OTEC power potential in the coarse-resolution CMIP6 CGCM ensemble mean (7.21 ± 0.30 TW) is significantly smaller than those in the CESM-H (8.55 ± 0.11 TW) and observation (9.36 ± 0.04 TW). Although the linear trend of OTEC power potential in the coarse-resolution CMIP6 CGCM ensemble mean during 1955-2021 (2.08 ± 0.31 TW/century) is larger than those in the CESM-H (1.99 ± 0.59 TW/century) and observation (1.80 ± 0.22 TW/century), the differences of the trends between the coarse-resolution CMIP6 CGCM ensemble mean, CESM-H and observation are not statistically significant. Therefore, we have to admit that we overstate the difference in the simulated OTEC power potential between the CMIP6 CGCMs and CESM-H. In the revised manuscript, we have revised the statement as:

“Based on the above comparisons, we conclude that the CESM-H generally provides a reliable simulation of OTEC power potential during 1955-2021. Specifically, it simulates a linear trend of OTEC power potential similar to those in the observation and CMIP6 CGCM ensemble mean but outperforms the CMIP6 CGCM ensemble mean in the simulated time-mean OTEC power potential. This lends support to using the CESM-H for projecting the future OTEC power potential change by the end of this century.” (Line 134-139)

Finally, we have shown the location of the EEZs across the globe in the revised Fig. 1f,g,h.

1. Line 17: I believe that the authors mean “power potential”.

Revised. Please see Line 14.

2. Line 19: “the exclusive economic zone” needs to be modified for clarity. Maybe the authors mean the exclusive economic zones across the globe?

Thank you for your comment. This phrase has been deleted due to the word limits of the abstract required by Nature Communications.

3. Line 21: In the North Atlantic?

The offsetting effect on the OTEC power potential increase due to deep ocean warming is strongest in the North Atlantic Ocean (Supplementary Figure 6b). Although the offsetting effect in the South Atlantic Ocean is less evident than that in the North Atlantic Ocean, it is still stronger than that in the Pacific and Indian Oceans. In the revised manuscript, we have revised the sentence as:

“The offsetting effect is more evident in the Atlantic Ocean than Pacific and Indian Oceans.” (Line 17)

4. Line 23: The authors should remove the phrase starting with “unresolved by the majority of current generation of climate models” as those models parameterize the impact of eddies the eddies. The additional factor that was not directly examined was the impact of a weaker mean flow in the control climates in the CMIP6 that could

explain the offset shows in Extended Data Figure 1. In addition, Extended Data Fig. 1 shows clearly that ΔT is reduced by a similar amount in the CMIP6 runs as that in CESM-H. This figure suggests that the increase in OTEC potential may be well represented by the parameterized eddies in the CMIP6 models even if the mean OTEC is not.

(a) Reply to your comment on removing the phrase starting with “unresolved by the majority of the current generation of climate models”

Thank you for pointing out this important issue. We have deleted this statement “unresolved by the majority of the current generation of climate models” in the revised manuscript.

(b) Reply to your comment on the effect of differed mean flows on the difference of simulated time-mean OTEC power potential between the CESM-H and coarse-resolution CMIP6 CGCMs

Griffies et al. (2015) compared the simulations between a high-resolution CGCM resolving mesoscale eddies and coarse-resolution CGCM parameterizing mesoscale eddies' effects using the GM90 parameterization (Gent & McWilliams, 1990). Similar to the finding of this study, they reported stronger thermal stratification in the high-resolution than coarse-resolution CGCM and attributed this thermal stratification difference to the difference between the effects of resolved and parameterized mesoscale eddies. In the high-resolution (coarse-resolution) CGCM, the mean flows generate a downward vertical heat transport (VHT) that is largely balanced by the upward VHT by resolved (parameterized) mesoscale eddies. It indicates that mean flows act to cool the sea surface but warm the deep ocean, reducing the thermal stratification, i.e., a destratification effect. The opposite is true for mesoscale eddies, i.e., a restratification effect. However, the parameterized upward VHT in the coarse-resolution CGCM is significantly weaker than the resolved VHT in the high-resolution CGCM (their Figure 12a and c). Accordingly, the downward VHT by mean flows also becomes weaker in the coarse-resolution CGCM than high-resolution CGCM to maintain an equilibrium state. As the destratification effect by mean flows is stronger in the high-resolution than coarse-resolution CGCM, mean flows cannot account for the enhanced time-mean thermal stratification in the CESM-H than coarse-resolution CMIP6 CGCMs. Instead, it is more likely to result from the stronger restratification effect by the resolved than parameterized mesoscale eddies.

The discrepancy between the VHT by resolved and parameterized mesoscale

eddies suggests deficiencies of the GM90 parameterization widely used in the coarse-resolution CGCMs. The deficiencies are likely to be multifaceted. In particular, the GM90 parameterization does not account for the VHT generated by mesoscale eddies via the turbulent thermal wind balance that is found to play an important role in the upper ocean (Jing et al., 2020).

(c) Reply to your comment on the effect of parameterized mesoscale eddies on the change of OTEC power potential under greenhouse warming

Although the linear trends of OTEC power potential between the CESM-H and coarse-resolution CMIP6 CGCM ensemble mean do not statistically differ from each other during the historical period (1955-2021), it does not necessarily mean that the CESM-H and coarse-resolution CMIP6 CGCMs have consistent responses of OTEC power potential to greenhouse warming. With the rising greenhouse gas emission in the future as in a high carbon emission scenario, the difference of projected linear trends of OTEC power potential between the CESM-H and coarse-resolution CMIP6 CGCM ensemble mean is qualitatively similar to its historical counterpart but quantitatively becomes sufficiently large to be statistically significant (Figure 1a). Therefore, it is likely that the different responses of OTEC power potential to greenhouse warming between the CESM-H and coarse-resolution CMIP6 CGCMs have not emerged by now due to the large natural variability in the simulated OTEC power potential. For this reason, it remains unclear whether the mesoscale eddy parameterization in the coarse-resolution CMIP6 CGCMs can well represent the change of mesoscale eddies and its effect on the OTEC power potential change under greenhouse warming.

We have added the above discussion to the revised manuscript. Please see Line 238-251.

5. Line 30: Please give an example of what is meant by “intermittent renewable technologies”

The phrase “intermittent renewable technologies” means renewable energy sources that cannot generate electricity steadily, such as wind and solar energy which relies on the availability of strong winds and sunlight. Table R1 compares the capacity factor (CF) of power plants for different types of energy sources (Garduño-Ruiz et al., 2021). The CF is defined as the ratio of actual electrical energy output over a given period to the theoretical maximum electrical energy output over that period and is often used to measure the steadiness of different energy sources. Among the common renewable

energy sources, the OTEC has the highest value of CF and is thus most stable. In the revised manuscript, we have cited Garduño-Ruiz et al. (2021) and revised this phrase as:

“Unlike many other renewable technologies based on intermittent energy sources such as winds and sunlight”. (Line 25-26)

Table R1 | Capacity factor (CF) and levelized cost of energy (LCOE) for different types of power plants. Data sources are from Garduño-Ruiz et al. (2021) (their figure 9).

Technologies	CP (%)	LCOE (USD/MWh)
OTEC	92	140-157
Advanced Nuclear	90	100
Biomass	82	105
Conventional Coal	84	97
Geothermal	91	99
Hydroelectric	52	70
Natural Gas	88	96
Solar Photovoltaic	25	210
Solar Thermal	18	310
Wind Offshore	34	245
Wind Onshore	34	80

6. Line 35: At some point in the paper, the large cost of implementation of OTEC should be mentioned when compared to other technologist.

Garduño-Ruiz et al. (2021) have compared the levelized cost of energy (LCOE) of power plants using different types of energy sources (Table R1). The LCOE measures the lifetime cost of a power plant divided by its net power production and is often used to compare the cost of electricity generation via different technologies. The LCOE of an OTEC plant lies between 140 and 157 USD/MWh and is moderate among power plants using various technologies. In the revised manuscript, we have cited Garduño-Ruiz et al. (2021) and stated the LCOE of an OTEC plant. Please see Line 28.

7. Line 41-42: I am not sure what the authors mean by “with even a larger increasing rate”? Do they mean because global warming is expected to accelerate? Doesn’t this

depend on the emission scenario considered?

We agree with you that the rate of strengthening of thermal stratification depends on the emission scenario. In the revised manuscript, we have deleted the phrase “with even a larger increasing rate” and revised the sentence as:

“In the future, the strengthening of thermal stratification is likely to continue due to greenhouse warming, implying enriched OTEC resources.” (Line 39-41)

8. Line 44: More explanation is needed here: do the authors think that contributions from ocean heat transport by the large scale flow is expected to change?

Yes, previous studies (Couldrey et al., 2021; Dias et al., 2020; Garuba & Klinger, 2016; Gregory et al., 2016; Wu et al., 2012; Zika et al., 2021) suggest that the heat transport by oceanic flows changes significantly under greenhouse warming and plays an important role in determining the anthropogenic change of ocean thermal structure. We have revised this part as:

“On the one hand, the SST changes caused by local sea surface heat flux changes can be advected elsewhere by oceanic flows like a passive tracer, particularly into the deep ocean via the ventilation processes (Church et al., 1991). On the other hand, changes in surface wind and buoyancy forcing under greenhouse warming drive changes in oceanic flows that redistribute the heat in the ocean and further affect the efficiency of ocean uptake of anthropogenic heat surplus via the redistribution feedback. (Gregory et al., 2016)” (Line 44-48)

9. Line 46: Please clarify what is meant by “more reliable”? Is this reference when compared against the 1-D models?

Yes, we have made this point clear in the revised manuscript. Please see Line 54.

10. Line 48: I assume you mean 1 degree (not 1 degree C).

Corrected. Thanks.

11. Line 49: I assume the authors mean the coastal ocean here. That should be said explicitly.

We have replaced the phrase “nearshore region” with “coastal ocean” following your advice. Please see Line 56.

12. Line 61: What is meant by “moderate cold-water intake”? This is not clear from the description of OTEC provided here.

Sorry for the unclarity. Here the moderate cold-water intake rate means 5 m/year, the value used for the computation of OTEC power potential in this study. We have made this point clear in the revised manuscript (See Line 68-69). Moreover, we have provided more explanations of the feedback effect on ocean thermal stratification caused by utilizing OTEC in the method section. Please see Line 336-341.

13. Line 63: This should read “potential power” not power as the study here gives an upper limit on what could be produced.

Thanks for your advice. It has been changed to “power potential” in Line 70-71 following your advice.

14. Line 75: Please replace “only about” with the mean and standard deviation.

Revised. Please see Line 85.

15. Line 79: the difference between the trends among the observation, CESM-H and CMIP6 appear to be small in Extended Data Fig. 1. Is this a statistically robust statement? If so, they should give the standard error in the trend estimate, including the spread across the CMIP6. Or remove this statement.

We are grateful to you for pointing out this important issue. The linear trends of OTEC power potential during 1955-2021 among the observation, CESM-H, and coarse-resolution CMIP6 CGCMs are not statistically different from each other. We have removed this statement from the revised manuscript.

16. Line 89: What is “the OTEC region”. Please define.

The OTEC requires the temperature difference between the surface warm water and deep cold water to exceed 20°C (Etemadi et al., 2011; Rau & Baird, 2018), which means that only part of the global ocean can be exploited for the OTEC. In this study, the OTEC region is defined as the region where the temperature difference exceeds 20°C. Outside the OTEC region, the OTEC power potential is zero by definition. We have provided the definition of the OTEC region in the method section of the revised manuscript. Please see Line 115-117 and 328-333.

17. Line 98: This is a spurious comment about the coarse resolution of the observations and should be removed unless the authors do further analysis. I also suspect that error analysis would show that they are not statistically different.

We are grateful to you for pointing out this issue. In the revised manuscript, we have added the standard error of OTEC power potential. The time-mean OTEC power potential in the EEZ is 4.87 ± 0.02 TW in the observation, significantly larger than 4.69 ± 0.04 TW in the CESM-H. To examine the effect of coarse resolution on the OTEC power potential in the EEZ, we remap the OTEC power potential density simulated by the CESM-H from the CESM-H's fine grids ($\sim 0.1^\circ$) to the coarse grids (1°) of the observation, using a bilinear interpolation. Figure R1 compares the time series of the CESM-H simulated OTEC power potential in the EEZ computed based on the two different grids. Their time-mean values are almost identical, differing by less than 0.01TW. So the difference of the OTEC power potential in the EEZ between the CESM-H and observation cannot be attributed to the coarse resolution of observational data. We have removed the inappropriate statement in the revised manuscript.

Figure R1 | Time series of the CESM-H simulated OTEC power potential in the EEZ computed based on the CESM-H's fine grids ($\sim 0.1^\circ$) and coarse grids (1°) of the observation.

18. Line 105: I see good agreement with the increase of the CMIP6 within the standard deviation. This sentence should be modified to reflect that even though the mean delta T is small, the increase is in relatively good agreement. This also requires that the paragraph starting at line be modified.

Thanks for your comment. The linear trends of ΔT during 1955-2021 among the observation, CESM-H and coarse-resolution CMIP6 CGCMs are not statistically different from each other. We have revised this paragraph accordingly. Please see Line 132-133.

19. Line 130 the statement starting with “Zoom” is not a sentence.

The sentence has been revised as:

“We then focus on the more geographically restrictive regions like the South China Sea (SCS) and Gulf of Mexico (GOM), the two representative marginal seas of the Pacific and Atlantic Oceans, respectively.” (Line 164-165)

20. Line 156: Sentence starting with “Sensitivity” does not make sense.

The water column, i.e., 800-1200 m, chosen for the ocean heat content (OHC) budget analysis is somewhat arbitrary. We have performed some analysis to test whether the major results of the budget analysis are sensitive to the choice of upper and lower bounds of the water column. It is found that the dominant role of vertical mesoscale eddy heat transport in the deep ocean warming also holds for the water column of 900-1100 m and 600-1400 m (Figure R2). In the revised manuscript, we have rewritten this sentence as:

“Changing the range of the water column for analysis to 900-1100 m or 600-1400 m does not have any substantial impact on the following results (Supplementary Fig. 7).” (Line 202-204)

Figure R2 | (a) The anomalous OHC budget under greenhouse warming (i.e., 2071-2100 minus 1992-2021) in the 800-1200 m water column over the Atlantic OTEC region where TD represents the OHC tendency, Q_{mean} the heat transport convergence of mean flows, Q_{eddy} the heat transport convergence by mesoscale eddies, and Q_{mix} the parameterized turbulent vertical mixing. (b), (c) Same as (a) but for the 900-1100 m and 600-1400 m water column, respectively. Contribution of vertical mesoscale eddy heat transport at 800 m, 900 m and 600 m to Q_{eddy} is marked by the hatched lines in (a),

(b) and (c), respectively.

21. Line 162: Once again, it is important to note that the effects of mesoscale processes are parameterized in the CMIP6 models.

Thanks for your comment. As mentioned in our reply to your comment#2, although the coarse-resolution CMIP6 CGCMs parameterize the mesoscale eddies, previous studies (Griffies et al., 2015; Jing et al., 2020) suggest that the parameterization has noticeable deficiencies. Furthermore, it remains unclear whether the mesoscale eddy parameterization in the coarse-resolution CMIP6 CGCMs can well represent the change of mesoscale eddies and its effect on the OTEC power potential change under greenhouse warming.

22. Line 175: It should be pointed out that this is likely owing to the shallower thermocline in the Pacific and Indian Oceans.

We are grateful to you for raising this explanation. According to Zhang et al. (2013), a shallower thermocline will make the vertical structure of mesoscale eddies shallower. So it is very likely that the shallower vertical mesoscale eddy heat transport in the Pacific and Indian Oceans than in the Atlantic Ocean is partially due to the shallower thermocline in the Pacific and Indian Oceans (Talley et al., 2011). We have mentioned this possible explanation and cited Talley et al. (2011) and Zhang et al. (2013) in the revised manuscript. Please see Line 222.

23. Line 182: It is important to note that while the low resolution CMIP6 simulations do not represent the mean OTEC power, the CHANGE in OTEC power is well represented.

This point has been explicitly stated in Line 89-93 and 132-133.

24. Line 184: I do understand what is meant by this sentence.

Sorry for the unclarity. The CESM-H, as well as CGCMs in CMIP6, does not simulate the feedback effect on ocean thermal stratification caused by the effluent discharge when utilizing OTEC. According to Rajagopalan and Nihous (2013) (see their Figure 2), this feedback effect becomes nonnegligible for the cold-water intake rate $w_{cw} > 5\text{m/year}$ and invalidates the linear dependence of OTEC power potential on w_{cw} . For this sake, in this study, we choose $w_{cw} = 5\text{m/year}$ under which condition the feedback is negligible. This sentence has been revised as:

“CESM-H, as well as CMIP6 CGCMs, does not simulate the feedback effect on ocean thermal stratification caused by the effluent discharge when utilizing OTEC. This limits us to assign a moderate cold-water intake rate under which condition the feedback is negligible.” (Line 251-254)

25. In the figure captions “OTEC power” should be replaced with OTEC power potential.

Revised.

26. Figure 3: the standard error of the trend estimates should be stated for each of the lines in the plots here.

The standard error has been added in Figure 3.

27. Line 262: Should be changed to “the ocean thermal stratification in practice”

We have revised this part. Please see Line 340.

28. Line 263: how big is 5m/year compared to typical values of w from mixing or upwelling? Some context is needed. 5m/year seems like a larger number for the mid-depth ocean.

A cold-water intake rate w_{cw} of 5m/year is on the same order of the large-scale vertical velocity in the mid-depth ocean interior estimated to be $O(1 \text{ m/year})$ (Liang et al., 2017; Munk & Wunsch, 1998) but is more than an order of magnitude smaller than the wind-driven upwelling $O(1 \text{ m/day})$ in the eastern boundary upwelling systems (Brady et al., 2017). We have added these reference values in the revised manuscript. Please see Line 341-344.

29. Line 274: Over what area would this hold? A few details of the sensitivity tests done should be included.

The area here is the Atlantic OTEC region (Supplementary Fig. 6b) (Line 200). The CESM-H saves the entire diagnostic output for the temperature governing equation during 1920-1934. This allows us to test whether the horizontal mixing averaged over the Atlantic OTEC region is negligible compared to the other terms. Figure R3 shows the time series of individual terms in the OHC budget for the 800-1200 m water column

over the Atlantic OTEC region. Both the time-mean value and variation of horizontal mixing are negligible compared to other terms in the budget, lending supports to our argument. In the revised manuscript, we have revised the sentence as:

“The horizontal mixing by subgrid-scale processes is dropped, as its effect is negligible compared to other terms when averaged over a sufficiently large area considered here, i.e., the Atlantic OTEC region (Supplementary Fig. 6b; Supplementary Fig. 9).” (Line 370-373)

Figure R3 | The time series of individual terms in the OHC budget for the 800-1200 m water column over the Atlantic OTEC region (Supplementary Fig. 6b). Here the OHC tendency (TD), heat transport convergence by mean flows Q_{mean} , heat transport convergence by mesoscale eddies Q_{eddy} , vertical mixing Q_{mix}^v and horizontal mixing Q_{mix}^h can be explicitly computed.

References

- Brady, R. X., Alexander, M. A., Lovenduski, N. S., & Rykaczewski, R. R. (2017). Emergent anthropogenic trends in California Current upwelling. *Geophysical Research Letters*, 44(10), 5044–5052. <https://doi.org/10.1002/2017GL072945>
- Church, J. A., Godfrey, J. S., Jackett, D. R., & McDougall, T. J. (1991). A Model of Sea Level Rise Caused by Ocean Thermal Expansion. *Journal of Climate*, 4(4), 438–456. [https://doi.org/10.1175/1520-0442\(1991\)004<0438:AMOSLR>2.0.CO;2](https://doi.org/10.1175/1520-0442(1991)004<0438:AMOSLR>2.0.CO;2)
- Couldrey, M. P., Gregory, J. M., Boeira Dias, F., Dobrohotoff, P., Domingues, C. M., Garuba, O., et al. (2021). What causes the spread of model projections of ocean

- dynamic sea-level change in response to greenhouse gas forcing? *Climate Dynamics*, 56(1), 155–187. <https://doi.org/10.1007/s00382-020-05471-4>
- Dias, F. B., Fiedler, R., Marsland, S. J., Domingues, C. M., Clément, L., Rintoul, S. R., et al. (2020). Ocean heat storage in response to changing ocean circulation processes. *Journal of Climate*, 33(21), 9065–9082. <https://doi.org/10.1175/JCLI-D-19-1016.1>
- Etemadi, A., Emdadi, A., AsefAfshar, O., & Emami, Y. (2011). Electricity generation by the ocean thermal energy. *Energy Procedia*, 12, 936–943. <https://doi.org/10.1016/j.egypro.2011.10.123>
- Garduño-Ruiz, E. P., Silva, R., Rodríguez-Cueto, Y., García-Huante, A., Olmedo-González, J., Martínez, M. L., et al. (2021). Criteria for Optimal Site Selection for Ocean Thermal Energy Conversion (OTEC) Plants in Mexico. *Energies*, 14(8), 2121. <https://doi.org/10.3390/en14082121>
- Garuba, O. A., & Klinger, B. A. (2016). Ocean Heat Uptake and Interbasin Transport of the Passive and Redistributive Components of Surface Heating. *Journal of Climate*, 29(20), 7507–7527. <https://doi.org/10.1175/JCLI-D-16-0138.1>
- Gent, P. R., & McWilliams, J. C. (1990). Isopycnal Mixing in Ocean Circulation Models. *Journal of Physical Oceanography*, 20(1), 150–155. [https://doi.org/10.1175/1520-0485\(1990\)020<0150:IMIOCM>2.0.CO;2](https://doi.org/10.1175/1520-0485(1990)020<0150:IMIOCM>2.0.CO;2)
- Gregory, J. M., Bouttes, N., Griffies, S. M., Haak, H., Hurlin, W. J., Jungclaus, J., et al. (2016). The Flux-Anomaly-Forced Model Intercomparison Project (FAFMIP) contribution to CMIP6: investigation of sea-level and ocean climate change in response to CO₂ forcing. *Geoscientific Model Development*, 9(11), 3993–4017. <https://doi.org/10.5194/gmd-9-3993-2016>
- Griffies, S. M., Winton, M., Anderson, W. G., Benson, R., Delworth, T. L., Dufour, C. O., et al. (2015). Impacts on ocean heat from transient mesoscale eddies in a hierarchy of climate models. *Journal of Climate*, 28(3), 952–977. <https://doi.org/10.1175/JCLI-D-14-00353.1>
- Jing, Z., Wang, S., Wu, L., Chang, P., Zhang, Q., Sun, B., et al. (2020). Maintenance of mid-latitude oceanic fronts by mesoscale eddies. *Science Advances*, 6(31),

eaba7880. <https://doi.org/10.1126/sciadv.aba7880>

- Liang, X., Spall, M., & Wunsch, C. (2017). Global Ocean Vertical Velocity From a Dynamically Consistent Ocean State Estimate. *Journal of Geophysical Research: Oceans*, 122(10), 8208–8224. <https://doi.org/10.1002/2017JC012985>
- Munk, W., & Wunsch, C. (1998). Abyssal recipes II: energetics of tidal and wind mixing. *Deep Sea Research Part I: Oceanographic Research Papers*, 45(12), 1977–2010. [https://doi.org/10.1016/S0967-0637\(98\)00070-3](https://doi.org/10.1016/S0967-0637(98)00070-3)
- Rajagopalan, K., & Nihous, G. C. (2013). An assessment of global Ocean thermal energy conversion resources with a high-resolution ocean general circulation model. *Journal of Energy Resources Technology*, 135(4), 041202. <https://doi.org/10.1115/1.4023868>
- Rau, G. H., & Baird, J. R. (2018). Negative-CO₂-emissions ocean thermal energy conversion. *Renewable and Sustainable Energy Reviews*, 95, 265–272. <https://doi.org/10.1016/j.rser.2018.07.027>
- Talley, L. D., Pickard, G. L., Emery, W. J., & Swift, J. H. (2011). Chapter 4 - Typical Distributions of Water Characteristics. In L. D. Talley, G. L. Pickard, W. J. Emery, & J. H. Swift (Eds.), *Descriptive Physical Oceanography (Sixth Edition)* (pp. 67–110). Boston: Academic Press. <https://doi.org/10.1016/B978-0-7506-4552-2.10004-6>
- Wu, L., Cai, W., Zhang, L., Nakamura, H., Timmermann, A., Joyce, T., et al. (2012). Enhanced warming over the global subtropical western boundary currents. *Nature Climate Change*, 2(3), 161–166. <https://doi.org/10.1038/nclimate1353>
- Zhang, Z., Zhang, Y., Wang, W., & Huang, R. X. (2013). Universal structure of mesoscale eddies in the ocean. *Geophysical Research Letters*, 40(14), 3677–3681. <https://doi.org/10.1002/grl.50736>
- Zika, J. D., Gregory, J. M., McDonagh, E. L., Marzocchi, A., & Clément, L. (2021). Recent Water Mass Changes Reveal Mechanisms of Ocean Warming. *Journal of Climate*, 34(9), 3461–3479. <https://doi.org/10.1175/JCLI-D-20-0355.1>

Reply to the second reviewer

We are very grateful to you for your time in carefully reading our manuscript and providing helpful comments that make our manuscript better. We have carefully considered each of your comments (in blue) and revised the manuscript accordingly. Please find our response (in black) to your comments below.

Reviewer #2 (Remarks to the Author):

The paper studies the influence of global warming on the increase in power generation of OTEC plants. Such work is useful for extracting thermal energy from the ocean. There are some concerns that the authors are encouraged to address.

1. The common range of temperature differences between the surface water and deep water in different regions of the world can be quantified in the paper. The thermocline in different regions and the reasons for differences in temperature difference with depth can be discussed to better explain the zonal differences in OTEC power. The steady power (Line 31) is mostly suitable for equatorial regions where the surface temperatures do not vary a lot with seasons. Large temperature differences mentioned in the abstract is also not very informative since OTEC operates on temperature differences of $\sim 20^{\circ}\text{C}$.

(a) Reply to your comment on the distribution of temperature difference and underlying dynamics

Following your comment, we have displayed and discussed the spatial distributions of SST (T_s), the deep ocean temperature at 1000 m (T_{1000}) and their difference ($\Delta T = T_s - T_{1000}$) as well as the underlying dynamics (Fig. R1 and Supplementary Fig. 2). In the observation, the value of ΔT generally ranges from 0°C to 25°C in the global ocean, making the OTEC only available over approximately half of the global ocean. The spatial distributions of ΔT and OTEC power potential density P_{net} are primarily attributed to that of the SST (Fig. R1a and c; Fig. 1b). As the SST decreases poleward due to the latitudinally varying solar radiation, a nonzero P_{net} is confined to the low-latitude regions between 35°S - 40°N . Furthermore, there is a notable zonal asymmetry in the P_{net} . In the tropics, the SST and P_{net} are higher in the Indo-Pacific warm pool than the Pacific and Atlantic equatorial cold tongues. The former is due to the accumulation of warm surface water by the wind-driven ocean

circulations, whereas the latter is due to the upwelling of cold water from the thermocline into the surface layer (Talley et al., 2011a). In the subtropical oceans, high values of SST and P_{net} are centered in the west of ocean basins caused by the wind-driven anticyclonic ocean circulations (Talley et al., 2011a). The value of SST is further decreased in the eastern boundary upwelling systems due to the intense upwelling driven by along-shore equatorward winds (García-Reyes et al., 2015), leading to zero P_{net} in these regions. The T_{1000} spatially varies to a less extent compared to the SST but plays a non-negligible role in the regional variability of ΔT and P_{net} (Fig. 1b and c; Fig. R1b). In particular, the injection of salty, warm Mediterranean Water into the deep Atlantic Ocean results in high value of T_{1000} in the eastern subtropical Atlantic Ocean (Richardson et al., 2000; Talley et al., 2011b), reducing the value of ΔT to below 20°C and making P_{net} become zero. Similarly, the relatively low values of ΔT and P_{net} in the Arabian Sea than in the adjacent ocean are due to the injection of salty, warm Red Sea Water (Beal et al., 2000). Please see Line 95-113 in the revised manuscript for the above discussion.

Figure R1 | The spatial distribution of time-mean SST (a), deep ocean temperature at 1000 m T_{1000} (b), and their difference ΔT (c) during 1992-2021 obtained from the observation. The black solid line denotes the zero contour of the time-mean OTEC power potential density P_{net} during 1992-2021.

(b) Reply to your comment on the seasonality of OTEC power potential

In this study, we focus on the long-term change of annual mean global OTEC power potential under greenhouse warming. As you pointed out, the OTEC power potential density P_{net} at some locations can vary significantly at seasonal time scales. Following your comment, we recompute the time mean and linear trend of OTEC power potential over the global ocean in different months, respectively (Table R1). The differences among months are not statistically different from each other. Therefore, the conclusion in Line 31 of our original manuscript is robust across all the seasons. This is likely because the seasonal variations in the northern and southern hemispheres largely cancel each other for the global OTEC power potential.

Table R1 | The time mean and linear trend of global OTEC power potential in different months derived from the CESM-H. The error bar denotes the standard error.

Month	Time-mean (TW)	Linear Trend (TW/century)	
	1955-2021	1955-2021	2022-2100
January	8.55 ± 0.11	1.96 ± 0.55	4.87 ± 0.42
February	8.53 ± 0.11	1.98 ± 0.57	4.88 ± 0.41
March	8.51 ± 0.11	2.01 ± 0.56	4.92 ± 0.36
April	8.54 ± 0.10	2.03 ± 0.54	4.87 ± 0.33
May	8.56 ± 0.10	2.01 ± 0.55	4.86 ± 0.34
June	8.53 ± 0.10	2.01 ± 0.52	4.77 ± 0.34
July	8.53 ± 0.10	1.94 ± 0.52	4.71 ± 0.34
August	8.53 ± 0.09	1.99 ± 0.49	4.67 ± 0.37
September	8.55 ± 0.09	2.02 ± 0.50	4.67 ± 0.38
October	8.59 ± 0.10	2.00 ± 0.51	4.69 ± 0.40
November	8.62 ± 0.10	1.96 ± 0.52	4.79 ± 0.40
December	8.62 ± 0.10	1.99 ± 0.53	4.85 ± 0.41
Annual Mean	8.55 ± 0.11	1.99 ± 0.59	4.80 ± 0.42

(c) Reply to your comment on the large temperature difference mentioned in the abstract

We have revised the sentence as:

“The concept of utilizing a large temperature difference ($>20^{\circ}\text{C}$) between the surface and deep seawater to generate electricity ” (Line 10)

2. It would be interesting to see in the introduction some specific global warming mechanisms that target changes in ocean water temperatures, and by how much ocean water has changed over the years. Trends in changes of ocean surface and deep water temperatures over the years would be useful - higher rate of change in temperatures is projected in Extended Fig. 1. The high carbon emission scenario leading to temperature changes of ocean water discussed in the paper can also be quantified.

Thanks for your comment. We have briefly discussed the major mechanisms via which greenhouse warming causes temperature changes in the ocean. These mechanisms suggest the important role of heat transport of oceanic flows in the spatial distribution of anthropogenic temperature changes in the ocean, highlighting the deficiencies of the 1-D model used by previous studies in evaluating the response of OTEC power potential to greenhouse warming and providing a better motivation for our study. In addition, we have reported the changes in the sea surface and deep ocean temperature in the past several decades. Please see Line 37-39 and 41-48.

Following your comment, we have shown the projected trends of sea surface temperature and deep ocean temperature under the high carbon emission scenario (Supplementary Fig. 6a). Both the sea surface and deep ocean exhibit significant warming under the high carbon emission scenario but the warming rate of the sea surface is much larger than that of the deep ocean.

3. It is easier to comprehend the influence of global warming on surface ocean water. Deep cold water used by OTEC plants is ~ 1000 m below the ocean surface and the surface atmospheric effects would be almost negligible. Also, there is a lack in natural mixing between surface warm water and deep cold water. Hence, it is not easy to follow how DOW would be directly impacted by global warming. More discussions are required, particularly on the influence of ocean currents which is a focus of this paper. DOW temperatures remain almost constant ($4 - 6^{\circ}\text{C}$) at 1000 m throughout the world, while surface temperatures vary with region. Therefore, some form of evidence is necessary that the DOW does change with global warming.

(a) Reply to your comment on the mechanisms causing deep ocean temperature change under greenhouse warming

Thanks for your question. As you pointed out, turbulent mixing alone is inefficient to warm the deep ocean. There are two major mechanisms that can cause deep ocean temperature change under greenhouse warming. On the one hand, the SST changes caused by local sea surface heat flux changes can be advected elsewhere by oceanic flows like a passive tracer, particularly into the deep ocean via the ventilation processes (Church et al., 1991). On the other hand, changes in surface wind and buoyancy forcing under greenhouse warming drive changes in oceanic flows that redistribute the heat throughout the ocean and further affect the efficiency of ocean uptake of anthropogenic heat surplus via the redistribution feedback (Gregory et al., 2016). We have discussed these mechanisms in the revised manuscript. Please see Line 41-48.

(b) Reply to your comment on the evidence of the deep ocean temperature change under greenhouse warming

Previous studies have revealed the significant warming of the deep ocean in the past several decades based on the observations, although it is much weaker than the warming of the sea surface (Cheng et al., 2017; Fox-Kemper et al., 2021). Fig. R2 shows the observational time series of global mean deep ocean temperature (1000 m) during 1955-2021 derived from the JMA data. Consistent with the previous studies, there is a significant positive trend. In the revised manuscript, we have cited the existing literature to provide evidence of deep ocean warming under anthropogenic climate changes. Please see Line 38.

Figure R2 | Time series of global mean deep ocean temperature at 1000 m derived from the JMA data. The thick black line shows the linear trend.

4. The different regions (Pacific, Atlantic, etc) can be labelled on the contour maps for better clarity in result presentation.

Thanks for your suggestion. We have labelled the different regions in Supplementary Fig. 6b.

5. It is difficult to link some of the discussions in the paper with the figures. This should be improved throughout the paper. (e.g. Line 114 – 115: refers to Figure 3a and Figure 1a?; Lines 120 – 122 – not clear; margin of OTEC?). Purpose of Table 1 is not clear.

Following your comments, we have made the discussion linked to the figures in the revised manuscript.

The margin of the OTEC region is defined as the region where the frequency of the temperature difference between the sea surface and deep ocean exceeding 20°C is less than 50%. We have provided its definition in the method section and marked it in the Supplementary Fig. 4 of the revised manuscript. Please see Line 152-154 and 333-335.

The purpose of Supplementary Table 1 is to list the CMIP6 CGCMs used in this study so that readers can reproduce our results.

6. The geographical differences between China and Mexico affects the OTEC power due to differences in vertical ocean thermal gradients. As mentioned above, reasons for such regional differences in thermal gradients and OTEC power should be well explained.

For the time-mean OTEC power potential (Fig. 1b), its difference between the SCS and GOM is primarily attributed to the difference in SST (Fig. R3). The SCS located at lower latitudes has higher SST than the GOM (Fig. R3a and b), causing a larger temperature difference ΔT between the sea surface and deep ocean at 1000 m and thus a larger OTEC power potential.

As to the projected OTEC power potential change under greenhouse warming (Fig. 2c and d), although both the SCS and GOM exhibit an evident increase in the OTEC power potential density P_{net} by the end of this century, their spatial structures of P_{net} changes differ substantially. The change in P_{net} from 1992-2021 to 2071-2100 is relatively homogenous in the SCS, being 23 kW/km² or so. In contrast, the change of P_{net} in the GOM is larger in the northern part (~40 kW/km²), but decreases

southeastward to $\sim 15 \text{ kW/km}^2$ near the Yucatan Channel. This heterogeneous change of P_{net} mimics that of SST change (Fig. R4). The depressed SST increase in the southeastern GOM is likely caused by the weakened Loop Current associated with a decline of Atlantic Meridional Overturning Circulation under greenhouse warming (Sen Gupta et al., 2021) that reduces the warm water intrusion into the GOM via the Yucatan Channel (Oey et al., 2013). Indeed, we find that the change of heat transport convergence by oceanic flows under greenhouse warming induces a strong cooling anomaly in the southeastern GOM (Fig. R5), contributing to the depressed SST increase there.

We have added the above discussion in the revised manuscript. Please see Line 164-176.

Figure R3 | Observed spatial distribution of time-mean SST (a), deep ocean temperature at 1000 m T_{1000} (b) and their difference ΔT (c) in the SCS during 1992-2021. (d)-(f) same as (a)-(c) but in the GOM.

Figure R4 | CESM-H simulated spatial distribution of time-mean SST (a), deep ocean temperature at 1000 m T_{1000} (b) and their difference ΔT (c) in the SCS during 2071-2100 minus their counterparts during 1992-2021. (d)-(f) same as (a)-(c) but in the GOM.

Figure R5 | CESM-H simulated spatial distribution of time-mean heat transport convergence of oceanic flows averaged over the upper 50 m during 2071-2100 minus its counterpart during 1992-2021.

7. As acknowledged, the OTEC power depends on the thermal gradient with depth, regardless of the rates of changes in the surface and deep ocean water. Therefore, global warming may not always improve OTEC power, since the temperature gradient could change or remain same. This could be highlighted as a key outcome and could serve as a guide for potential future researchers investigating this topic. The growth of OTEC resource mentioned in the title may not be appropriate as well.

Thanks for your invaluable advice. Although the surface ocean warms faster than the deep ocean in terms of the global average (Cheng et al., 2017; Fox-Kemper et al., 2021), the local change of the difference between the sea surface and deep ocean temperature ΔT under greenhouse warming is controlled by complicated dynamics and can differ substantially from its global mean value. As shown in Fig. R6a, some high-latitude regions such as the subpolar North Atlantic Ocean do show a decreased ΔT under greenhouse warming. However, these regions are outside the OTEC region. In the OTEC region, there is a universal increase of ΔT under greenhouse warming (Fig. R6a). Nevertheless, the change of ΔT can be locally much smaller than that caused by SST change alone (Fig. R6a and b), suggesting the important role of deep ocean warming in retarding the ΔT increase under greenhouse warming. In the revised

manuscript, we have highlighted this finding as a key outcome of the paper. Please see Line 228-232.

As the increase of ΔT under greenhouse warming holds in the OTEC region, we think the expression “the growth of OTEC resources under greenhouse warming” in the title is appropriate.

Figure R6 | (a) Spatial distribution of the time-mean ΔT during 2071-2100 minus that during 1992-2021 derived from the CESM-H. (b) Same as (a) but caused by SST change alone. The black solid line denotes the zero contour of the time-mean OTEC power potential density P_{net} during 2071-2100.

References

- Beal, L. M., Ffield, A., & Gordon, A. L. (2000). Spreading of Red Sea overflow waters in the Indian Ocean. *Journal of Geophysical Research: Oceans*, 105(C4), 8549–8564. <https://doi.org/10.1029/1999JC900306>
- Cheng, L., Trenberth, K. E., Fasullo, J., Boyer, T., Abraham, J., & Zhu, J. (2017). Improved estimates of ocean heat content from 1960 to 2015. *Science Advances*, 3(3), e1601545. <https://doi.org/10.1126/sciadv.1601545>

- Church, J. A., Godfrey, J. S., Jackett, D. R., & McDougall, T. J. (1991). A Model of Sea Level Rise Caused by Ocean Thermal Expansion. *Journal of Climate*, 4(4), 438–456. [https://doi.org/10.1175/1520-0442\(1991\)004<0438:AMOSLR>2.0.CO;2](https://doi.org/10.1175/1520-0442(1991)004<0438:AMOSLR>2.0.CO;2)
- Fox-Kemper, B., Hewitt, H. T., Xiao, C., Aðalgeirsdóttir, G., Drijfhout, S. S., Edwards, T. L., et al. (2021). Ocean, cryosphere, and sea level change. In V. Masson-Delmotte, P. Zhai, A. Pirani, S. L. Connors, C. Péan, S. Berger, et al. (Eds.), *Climate Change 2021: The Physical Science Basis. Contribution of Working Group I to the Sixth Assessment Report of the Intergovernmental Panel on Climate Change* (Vol. 9, pp. 1211–1362). Cambridge, United Kingdom and New York, NY, USA: Cambridge University Press. <https://doi.org/10.1017/9781009157896.001>
- García-Reyes, M., Sydeman, W. J., Schoeman, D. S., Rykaczewski, R. R., Black, B. A., Smit, A. J., & Bograd, S. J. (2015). Under Pressure: Climate Change, Upwelling, and Eastern Boundary Upwelling Ecosystems. *Frontiers in Marine Science*, 2. Retrieved from <https://www.frontiersin.org/articles/10.3389/fmars.2015.00109>
- Gregory, J. M., Bouttes, N., Griffies, S. M., Haak, H., Hurlin, W. J., Jungclaus, J., et al. (2016). The Flux-Anomaly-Forced Model Intercomparison Project (FAFMIP) contribution to CMIP6: investigation of sea-level and ocean climate change in response to CO₂ forcing. *Geoscientific Model Development*, 9(11), 3993–4017. <https://doi.org/10.5194/gmd-9-3993-2016>
- Oey, L.-Y., Ezer, T., & Lee, H.-C. (2013). Loop Current, Rings and Related Circulation in the Gulf of Mexico: A Review of Numerical Models and Future Challenges. In W. Sturges & A. Lugo-Fernandez (Eds.), *Geophysical Monograph Series* (pp. 31–56). Washington, D. C.: American Geophysical Union. <https://doi.org/10.1029/161GM04>
- Richardson, P. L., Bower, A. S., & Zenk, W. (2000). A census of Meddies tracked by floats. *Progress in Oceanography*, 45(2), 209–250. [https://doi.org/10.1016/S0079-6611\(99\)00053-1](https://doi.org/10.1016/S0079-6611(99)00053-1)
- Sen Gupta, A., Stellema, A., Pontes, G. M., Taschetto, A. S., Vergés, A., & Rossi, V. (2021). Future changes to the upper ocean Western Boundary Currents across two generations of climate models. *Scientific Reports*, 11(1), 9538.

<https://doi.org/10.1038/s41598-021-88934-w>

Talley, L. D., Pickard, G. L., Emery, W. J., & Swift, J. H. (2011a). Chapter 4 - Typical Distributions of Water Characteristics. In L. D. Talley, G. L. Pickard, W. J. Emery, & J. H. Swift (Eds.), *Descriptive Physical Oceanography (Sixth Edition)* (pp. 67–110). Boston: Academic Press. <https://doi.org/10.1016/B978-0-7506-4552-2.10004-6>

Talley, L. D., Pickard, G. L., Emery, W. J., & Swift, J. H. (2011b). Chapter 9 - Atlantic Ocean. In L. D. Talley, G. L. Pickard, W. J. Emery, & J. H. Swift (Eds.), *Descriptive Physical Oceanography (Sixth Edition)* (pp. 245–301). Boston: Academic Press. <https://doi.org/10.1016/B978-0-7506-4552-2.10009-5>

REVIEWERS' COMMENTS

Reviewer #1 (Remarks to the Author):

The authors have addressed all of my comments. The paper was a pleasure to read and I think it is ready to be accepted.

Reviewer #2 (Remarks to the Author):

All the comments have been satisfactorily addressed.